**communications** engineering

# Implantable silicon neural probes with nanophotonic phased arrays for single-lobe beam steering
Fu-Der Chen [1,2,3,8] ✉, Ankita Sharma [1,2,3,8] ✉, Tianyuan Xue[1,2], Youngho Jung[1], Alperen Govdeli[1,2], Jason C. C. Mak[2], Homeira Moradi Chameh [4], Mandana Movahed[4], Michael G. K. Brunk[1,3], Xianshu Luo[5], Hongyao Chua[5], Patrick Guo-Qiang Lo[5], Taufik A. Valiante [2,3,4,6,7], Wesley D. Sacher[1,3] & Joyce K. S. Poon [1,2,3] ✉

In brain activity mapping with optogenetics, patterned illumination is crucial for targeted neural stimulation. However, due to optical scattering in brain tissue, light-emitting implants are needed to bring patterned illumination to deep brain regions. A promising solution is silicon neural probes with integrated nanophotonic circuits that form tailored beam patterns without lenses. Here we propose neural probes with grating-based light emitters that generate a single steerable beam. The light emitters, optimized for blue or amber light, combine end-fire optical phased arrays with slab gratings to suppress higher-order sidelobes. *In vivo* experiments in mice demonstrated that the optical phased array provided sufficient power for optogenetic stimulation. While beam steering performance in tissue reveals challenges, including beam broadening from scattering and the need for a wider steering range, this proof-of-concept demonstration illustrates the design principles for realizing compact optical phased arrays capable of continuous single-beam scanning, laying the groundwork for advancing optical phased arrays toward targeted optogenetic stimulation.

Optogenetics with patterned photostimulation combines spatially targeted optical actuation of neurons with cell-type specificity to enable the investigation of synaptic connectivity in neuronal circuits at cellular resolutions. Discrete optical beam scanners, such as galvanometers and acousto-optic deflectors[1–3], spatial light modulators[4,5], and digital micromirror devices[6] are commonly used to impart patterns on an optical beam for spatially targeted photostimulation. However, these components form illumination patterns in free space, and the penetration depth of visible to near-infrared light (400–1100 nm) is limited to only about 1 mm from the brain surface in rodents[7]. To bring light into deep brain regions, implantable optical devices are being investigated, including optical fibers[8–10], miniature gradient index (GRIN) lenses[11–14], and silicon (Si) neural probes with micro light-emitting diodes ($\mu$LEDs)[15–18], organic LEDs[19], or integrated nanophotonic waveguides[20–25].

Among these options, nanophotonic waveguide-based Si neural probes are promising for delivering light with spatial precision. A rich

variety of light emission patterns can be achieved using waveguide-based grating emitters on the implant without the need for any lenses or discrete optics. In contrast, light-emitting implants, such as single-core optical fibers without wavefront compensation and LED probes, emit diffracting Gaussian and Lambertian beam profiles, respectively. Furthermore, nanophotonic waveguide probes have small form factors with widths and thicknesses ≤100 μm and sharp tips, which ease surgical implantation and displace less tissue compared to fiber bundles and GRIN lenses, which have typical diameters exceeding 300 μm. Through the design of grating emitters, we have demonstrated a wide range of beam patterns for optogenetic applications in the visible spectrum, including low-divergence beams[24,26], light sheets[26,27], focused[28], and steerable beams[29].

Steerable beams provide dynamically patterned illumination by serially scanning a beam across a continuous region in tissue samples. This approach is valuable for localized spatial mapping of neuronal connections within a circuit[1,2]. Optical phased arrays (OPAs) are common photonic

[1]Max Planck Institute of Microstructure Physics, Halle, Germany. [2]Department of Electrical and Computer Engineering, University of Toronto, Toronto, ON, Canada. [3]Max Planck-University of Toronto Centre for Neural Science and Technology, Toronto, ON, Canada. [4]Krembil Brain Institute, University Health Network, Toronto, ON, Canada. [5]Advanced Micro Foundry Pte. Ltd., Singapore Science Park II, Singapore. [6]Division of Neurosurgery, Department of Surgery, Toronto Western Hospital, University of Toronto, Toronto, ON, Canada. [7]Institute of Biomedical Engineering, University of Toronto, Toronto, ON, Canada. [8]These authors contributed equally: Fu-Der Chen, Ankita Sharma. ✉e-mail: fuder.chen@mail.utoronto.ca; ank.sharma@mail.utoronto.ca; joyce.poon@utoronto.ca

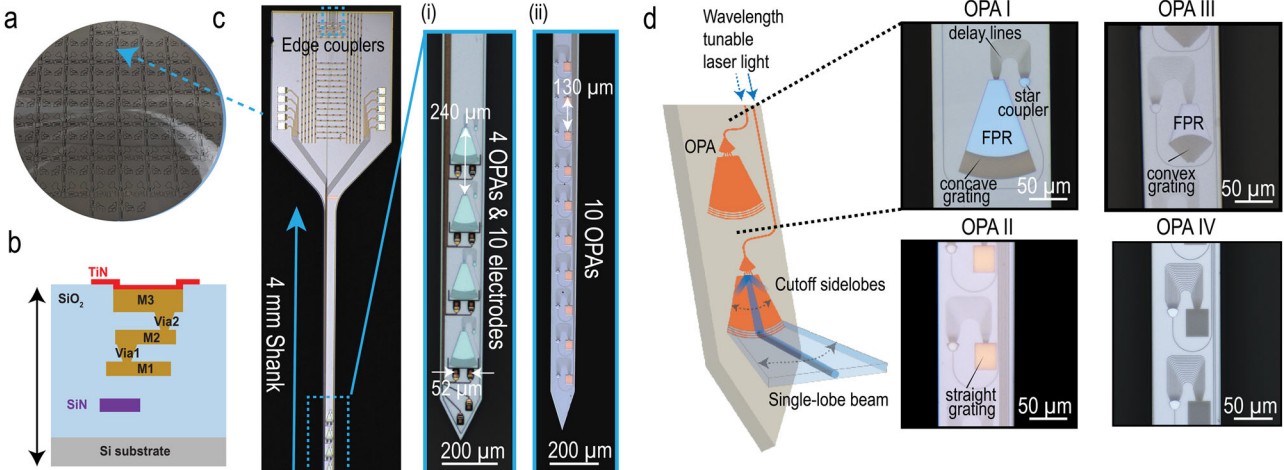

**Fig. 1 | Foundry-fabricated neural probes with single-lobe beam-steering OPA. a** 200-mm diameter Si wafers containing silicon neural probes. **b** Cross-sectional view of the wafer in (**a**). **c** Photograph of a beam-steering neural probe with inset (i) showing an optical micrograph of the 154 μm-wide shank containing 4 OPA emitters and 10 microelectrodes. The pitch between the OPAs is 240 μm and the horizontal pitch between the electrodes is 52 μm. Inset (ii) optical micrograph of a photonics-only shank variant containing 10 OPA emitters at a pitch of 130 μm on a shank with a width of 100 μm. The photonic devices span roughly 1.3 mm along the shank. **d** Illustration of the single-lobe OPA operation and annotated micrographs of four OPA designs with three of them having different slab grating curvatures. OPA Type IV shares a similar design to OPA Type II but is optimized for amber wavelengths. For all OPA designs, an end-fire array coupled to a slab grating enables the emission of a single steerable beam.

circuits to realize steerable optical beams without mechanically moving parts[29–33]. Notably, ref. 29 has demonstrated passive OPAs where the beam emission angle can be controlled by tuning the input wavelength, with the beam width matching near cell diameters(<23 μm full width at half maximum (FWHM) beam width at 50–150 μm propagation distance). However, one main limitation of the previous OPA design was the emission of multiple beams (grating orders) due to the emitter pitch exceeding the half-wavelength criterion for single-beam emission[31], compromising the spatial precision of the photostimulation. While it is possible to emit a single steerable beam at visible wavelengths without satisfying the $\lambda/2$ pitch criterion, such as by using microcantilevers[34], this approach is not suitable for implantable silicon neural probes because of its size and the need for mechanical actuators, which can damage the brain tissue.

Here, we overcome the challenge of scanning a single-lobe beam at depth in brain tissue with passive OPAs that combine end-fire phased arrays with slab gratings on Si neural probes. The slab grating design consists of an optional free-propagation region (FPR) for beam formation in the waveguide followed by a grating to radiate the beam out of the plane. This OPA circuit achieves single-beam emission due to (1) an in-plane emitter pitch of about $\lambda$, made possible by the short waveguides in the end-fire phased array, and (2) the slab grating that cuts off higher-angle sidelobes. We study the steering ranges in slab grating designs with concave, straight, and convex curvatures that operate in the blue or amber wavelength ranges. To demonstrate that these novel OPAs emit sufficient optical power, probes with sidelobe-free OPAs and integrated titanium nitride (TiN) electrodes, developed on the same multimodal neural probe platform reported in ref. 26, were used for photostimulation and recording in in vivo awake head-fixed animal experiments. These compact single-lobe beam-steering OPAs in the visible spectrum on implantable neural probes emit dynamic patterned illumination over a continuous region, a key step toward precise spatial mapping of neuronal activity in deep brain regions.

## Results
### OPA neural probes on 200-mm silicon wafers
The OPA neural probes were fabricated on 200-mm diameter Si wafers (see Fig. 1a) at Advanced Micro Foundry (AMF) using the process reported in refs. 26,35 and described in the "Methods" section. A cross-sectional view of these wafers is depicted in Fig. 1b. Waveguides were patterned in silicon nitride (SiN) formed by plasma-enhanced chemical vapor deposition (PECVD) or low-pressure chemical vapor deposition (LPCVD). To facilitate electrophysiological recordings, the neural probe platform included biocompatible titanium nitride (TiN) deposited on the surface for microelectrodes and three aluminum metal layers for routing. The vertical separation distance between the first aluminum layer (M1) and the SiN waveguide was set to >1 μm to achieve negligible metal-induced optical absorption loss at blue wavelengths (<0.02 dB/cm estimated in simulations for a 120 nm LPCVD SiN waveguide with a width of 300 nm).

A beam-steering neural probe is shown in Fig. 1c. Input laser light was coupled from a custom multicore fiber to the chip using edge couplers[24]. The probes operated in the blue (450–490 nm) or amber (575–600 nm) part of the spectrum. The nanophotonic circuits for blue light-emitting probes were defined in a 150 nm thick PECVD SiN waveguide layer (refractive index between 1.82 and 1.9 at 488 nm[24]) or 120 nm thick LPCVD SiN waveguide layer (refractive index of 2.04 at 488 nm[36]). Neural probes for the amber wavelengths used a 200 nm thick LPCVD SiN layer. Figure 1c inset (i) shows a hybrid neural probe design, which integrated four OPAs at a pitch of 240 μm and ten $20 \times 20\ \mu m^2$ electrodes on a 154 μm-wide shank and (ii) shows a photonic-only probe design, which had ten OPAs at a pitch of 130 μm spanning approximately 1.3 mm along a 100 μm-wide shank.

We investigated four different sidelobe-free OPA designs integrated into implantable neural probes. The specifications of these designs are summarized in Table 1. The OPA designs were passive to minimize the risk of tissue heating. Each OPA consisted of a star coupler, which divided the light from an input waveguide into 16 delay line waveguides. The delay lines had a differential path length of 5–6 μm to realize OPAs with a wide free spectral range (FSR) of ~ 20 nm. This design choice reduces the ratio of our supercontinuum laser source linewidth ( ~1–2 nm) to the FSR of the OPA, thereby minimizing beam broadening caused by the linewidth of the laser source. The delay lines converged to form an end-fire phased array, which emitted light in the plane of the probe. The short length of the end-fire waveguide array minimized the inter-waveguide crosstalk at an emitter pitch of about $\lambda$. The interference of the phased array emission to steer the beam primarily occurred within the chip, and the formed beam was radiated out of the plane by a slab grating. The input wavelength to the device tuned the in-plane emission angle, which also steered the out-of-plane beam. The

## Table 1 | Specifications of design parameters for different OPA types

| OPA | $\lambda$ (nm) | SiN Layer | $d$ (nm) | FSR (nm) | Slab Curvature | FPR Length (µm) | $\Lambda$ (nm) |
|---|---|---|---|---|---|---|---|
| I | 460–484 | 120 nm LPCVD | 400 | 24 | Concave | 100 | 480 |
| II | 450–468 | 150 nm PECVD | 400 | 19 | Straight | No FPR | 400 |
| III | 460–478 | 150 nm PECVD | 400 | 19 | Convex | 40 | 480 |
| IV | 574–598 | 200 nm LPCVD | 500 | 25 | Straight | No FPR | 670 |

$d$ is the pitch between waveguide emitters in the end-fire array.
FSR is the free spectral range.
FPR is the free-propagation region in the slab grating.
$\Lambda$ is the period of the slab grating.
A schematic annotating the parameters $d$ and $\Lambda$ can be found in Supplementary Note 2.

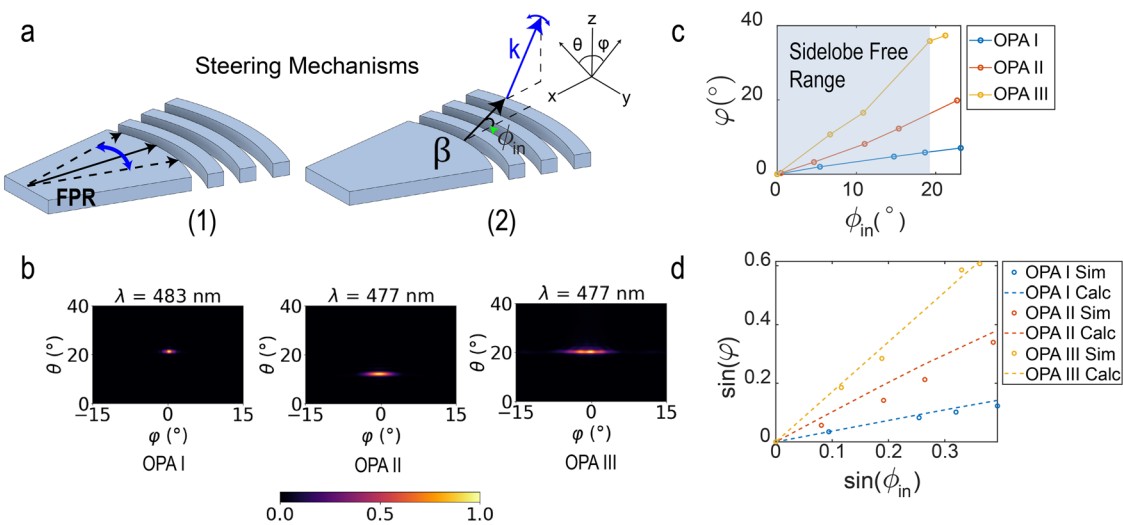

**Fig. 2 | Working principles of the single-lobe beam-steering OPA. a** Schematics of the two different steering mechanisms: (1) lateral translation of a beam in the free-propagation region (FPR), and (2) the dependence of the out-of-plane emission angle on the incident angle of the emission in the grating. (2) shows the directions of the far-field angles, $(\theta, \varphi)$ and in-plane angle $\phi_{in}$. $\beta$ is the propagation constant of the in-plane emission, $G$ is the grating vector and $k$ is the propagation constant of the output beam. **b** Simulated 3D finite-difference time-domain (FDTD) far-field beam profiles for each type of slab grating curvature. The far-field simulations are performed in a medium with a refractive index of 1.46, matching that of SiO2. **c** Comparison of the simulated steering ranges for the different types of slab grating curvatures. The circles are simulated data points, while the lines are interpolated. The shaded region indicates the sidelobe-free steering range. **d** The simulated relationship between $\varphi$ and $\phi_{in}$ is compared with the calculated relationship using Eq. (2a).

required wavelength range to complete a full steering cycle is centered close to the peak excitation wavelengths of the targeted opsins (ChR2 for blue designs and Chrimson for the amber design). This range can homogeneously excite the opsins with activation strength >70% due to the broad absorption band of these opsins[37].

Three of the four OPAs (I-III) were designed to operate at blue wavelengths with different slab grating curvatures as shown in Fig. 1d to tune the steering range in the far-field (see "OPA design" section for more details). For OPA I and III, light from the end-fired phased array emitted into a free-propagation region (FPR) and was subsequently radiated out of the plane of the probe with a slab grating at the end of the FPR. The FPR was introduced to transition between the end-fire phased array and the curved grating. Despite the phased array pitch not satisfying the $\lambda/2$ criterion, the high-order lobes of the end-fired phased array were cropped by the side walls of the FPR as illustrated in Fig. 1d. The FPR also improved the steering range of the device, as the OPA designs with the FPR achieved beam steering through two mechanisms illustrated in Fig. 2a: (1) the lateral translation of the beam that first formed in the FPR, and (2) the dependence of the out-of-plane emission angle on the incident angle of the FPR beam on the curved grating. For OPA II, with the aim of a reduced device footprint, the end-fired phased array was coupled directly to a straight one-dimensional slab grating. In this

design, high-order lobes with sufficiently large emission angles were minimized because they could not propagate along the full length of the slab grating. The design of OPA IV is similar to that of OPA II but was designed to operate in amber wavelengths to demonstrate the applicability of the OPA architecture in other wavelengths used in optogenetics. The four slab gratings had a constant grating period, a duty cycle of 50%, and were fully etched.

### OPA design

To intuitively understand the OPA operation, we first make the simplification to assume the optical input and output are plane waves. In this case, the relationship between the in-plane propagation angle, $\phi_{in}$, and out-of-plane emission angles, $(\theta, \varphi)$, follows the phase-matching condition,

$$\beta \cos \phi_{in} = k \sin \theta + m_x G_x, \quad (1a)$$

$$\beta \sin \phi_{in} = k \sin \varphi + m_y G_y, \quad (1b)$$

where $\beta$ is the propagation constant of the in-plane emission, $m_{x(y)}$ is an integer, $G_{x(y)}$ is the $x(y)$ component of the grating vector, and $k$ is the propagation constant of the output beam as depicted in Fig. 2a. For the first order beam emission, by setting $m = 1$, we can simplify the $y$ component to

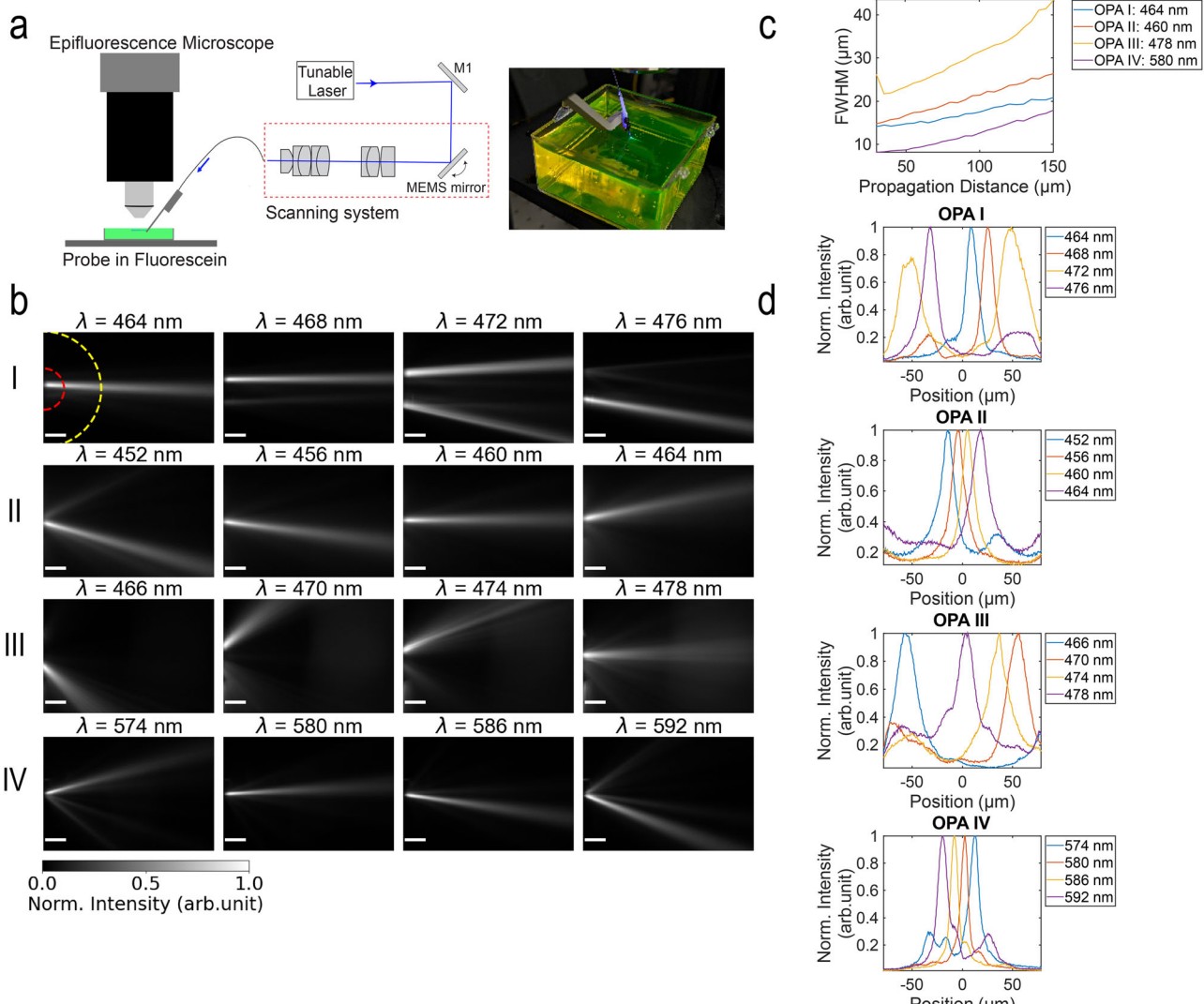

**Fig. 3 | Beam profile characterization in fluorescein. a** Diagram of the measurement setup showing a neural probe connected to a scanning system and a photograph of a probe emitting blue light in a 100 μM fluorescein fluorescent dye solution. **b** Measured top-down intensity beam profiles of the four OPA designs in solution at various wavelengths (λ). The scale bars are 50 μm. For each OPA, we characterized beam steering along an arc of 50 μm radius (annotated in red) and 140 μm radius (annotated in yellow). **c** Top-down measured FWHM beam widths versus propagation distance for the four OPA designs at a single wavelength. **d** Radial line profiles of the images in (**b**) at a propagation distance of 50 μm from the OPA. The line profiles are plotted as the intensity versus lateral position of the steerable beams along the arc with a 50 μm radius.

be

$$\sin \varphi = \Gamma \sin \phi_{in}, \qquad (2a)$$

$$\Gamma = \begin{cases} \frac{1}{k}\left(\beta - \frac{2\pi}{\Lambda}\right), & \text{for OPA Type I} \\ \frac{\beta}{k}, & \text{for OPA Type II} \\ \frac{1}{k}\left(\beta + \frac{2\pi}{\Lambda}\right), & \text{for OPA Type III} \end{cases} \qquad (2b)$$

where $\Gamma$ depends on the curvature of the slab grating.

We verified the OPA designs with three-dimensional (3D) finite-difference time-domain (FDTD) simulations. Examples of the simulated far-field beam profiles for each OPA are shown in Fig. 2b. These simulations confirmed the efficacy of the slab grating designs with the end-fire phased array in eliminating higher-order emissions. As the input wavelength is tuned, emitted beams are steered in $\varphi$, with small changes of <2.5° in $\theta$. A comparison of the simulation results, highlighting the steering capabilities of the three OPA designs in $\varphi$ for a given $\phi_{in}$, is shown in Fig. 2c. There is a trade-off between the beam width and the steering range. While a convex grating curvature has the

largest steering range, the width of the beam is also magnified, resulting in approximately the same number of resolvable points in the far-field for the three OPA designs. Figure 2d shows that the simulated relationship between $\phi_{in}$ and $\varphi$, agrees well with the values calculated using Eq. (2a).

### Beam profile characterization

To characterize the optical beam and steering properties of the OPAs, we measured the emission profiles of the fabricated neural probes in non-scattering media. Each probe was inserted into a chamber of fluorescent dye solution corresponding to its operating wavelength (450–484 nm: fluorescein or 574–598 nm: Texas Red). When immersed in solution, the probes were angled such that the emitted beams were approximately parallel to the surface of the liquid. The fluorescence signal was captured with an epifluorescent microscope illustrated in Fig. 3a. For the optical input, a laser scanning system similar to the one described in refs. 26,35,38 was used to address light to individual OPAs via a custom multicore fiber (MCF) attached to the probe[39]. Before attaching the MCF to the chip, we measured on-chip insertion losses between −18.5 dB and −22 dB, with a major portion of this loss (~8–11 dB) attributed to low edge coupling efficiency[24].

**Table 2 | Comparison of the OPA designs in fluorescein solution**

| OPA | λ (nm) | Device Size (mm²) | $L_{prop}$ = 50 μm | | $L_{prop}$ = 140 μm | |
|---|---|---|---|---|---|---|
| | | | Single-lobe Steering Arc Length (μm) | Beam Width FWHM (μm) | Single-lobe Steering Arc Length (μm) | Beam Width FWHM (μm) |
| I | 460–484 | 0.018 | 66.4 | 14.4 ± 2.5* | 79.8 | 17.5 ± 2.3* |
| II | 450–468 | 0.007* | 32.9 | 19.7 ± 3.1 | 78.2 | 29.7 ± 4.3 |
| III | 460–478 | 0.011 | 113.0* | 24.5 ± 6.6 | 235.4* | 44.0 ± 14.3 |
| IV | 574–598 | 0.006 | 33.0 | 9.9 ± 1.4 | 86.7 | 18.1 ± 2.4 |

*The best values for blue wavelengths.

After fiber attachment to the neural probe, additional losses ranging from ~2–12 dB were observed, depending on the MCF-to-edge coupler alignment after packaging. With an input power of 250 μW into the probe, the devices under test emitted approximately 0.2–2.5 μW. More details of the neural probe packaging and measurement setup are described in the "Methods" section.

A photograph of a packaged neural probe with an OPA emitting blue light in fluorescein solution is shown in Fig. 3a. The top-down fluorescence images of the beam profiles for the OPA designs at multiple wavelengths are shown in Fig. 3b. Each OPA was tuned across its full free spectral range (FSR) (close to 20 nm) by using a supercontinuum laser coupled to a tunable bandpass filter. The tunable laser source has a response time of 55 ms with a 1 nm step size, resulting in a tuning speed of approximately 1.1 s across the full FSR. Although this scanning speed may be sufficient for some neural mapping experiments[1], experiments requiring higher temporal precision[22,40,41] could benefit from replacing the mechanically moving diffraction grating in the tunable bandpass filter with an acousto-optic filter to enable sub-millisecond wavelength switching[20,42]. While Fig. 3 illustrates only a selection of wavelengths, the emitted beams can be continuously steered throughout the entire FSR, as demonstrated in Supplementary Movies 1 and 2.

The beam profiles measured in fluorescent dyes, shown in Fig. 3b, capture changes in emission angle along the φ-axis as well as additional lateral movements in the central position of the beam due to propagation in an FPR. Table 2 summarizes the device footprint and the steering results of the four OPA devices in fluorescent dye. In the context of simultaneous optogenetic stimulation and electrophysiological recording, we report the beam-steering range along arcs with a radius of 50 μm and 140 μm centered on the emitting region of the OPA. These propagation distances correspond to the estimated electrode detection range for large amplitude spikes (>60 μV) and spike amplitudes above background noise[43]. The single-lobe steering ranges reported in Table 2 have a peak beam intensity-to-background ratio between 3 and 20 times ($\gtrsim$4 dB). This background suppression level is sufficient to reliably induce spikes in the areas targeted by the main beam on neurons expressing ChR2 with a spiking probability ~70% while minimizing the light-induced spiking probability in untargeted regions to <35%[44]. The single-lobe steering range for the four OPA designs covers at least 60% of the total beam-steering range of the OPA (72%, 63%, 87%, and 81% for OPA I–IV, respectively).

When comparing steering performance among OPAs designed for blue wavelengths, OPA Type I stands out with the lowest beam divergence and the narrowest beam width of approximately 17.5 μm at a 140 μm propagation distance, as shown in Fig. 3c. In contrast, OPA Type III shows the widest single-lobe steering range enhanced by the negative curvature of the grating slab, as shown in Fig. 3d and reported in Table 2. Assuming that the length along a 180° arc represents the maximum steering range, the single-lobe steering range of OPA Type III covers 71% and 53% of the maximum range at propagation distances of 50 μm and 140 μm, respectively. Regarding the number of resolvable points (the single-lobe steering range/FWHM beam width), both OPA Types I and III achieved approximately 4–5 resolvable points over the single-lobe steering range, while OPA Type II had the lowest spatial resolution near the probe. Although OPA

Type II has the worst steering performance near the probe due to the lack of FPR, the design is advantageous for neural probes with dense OPA emitters, as it has the most compact footprint. Furthermore, OPA Type II can achieve larger steering ranges than OPA Type I at longer propagation distances ≥150 μm, resulting from the predominant influence of the grating curvature on beam steering (see Supplementary Note 1).

It should be noted that Fig. 3d presents normalized radial line cuts, and the maximum observed optical intensity varied with different excitation wavelengths due to several factors: the envelope function of the waveguide emitters in the end-fire phased array[45], efficiency variations in the edge couplers and slab gratings[24], and the excitation spectra of the dyes used for characterization[46]. Due to these combined effects, we observed maximum intensity variations of 8.1 dB, 4.5 dB, 4.3 dB, and 4.1 dB across one FSR for OPAs I–IV, respectively. In in vivo experiments, intensity variations can be compensated by adjusting the laser input power to the OPA device.

We also characterized beam steering in fixed tissue. We implanted two of the probes, one with OPA Type I for blue light and the other with OPA Type IV for amber light, into fixed brain slices from wild-type mice that were 2 mm thick and stained with the corresponding fluorescent dye. The probes were inserted at shallow depths such that the emitting OPAs were implanted <100 μm from the tissue surface. Figure 4 shows the beams formed in the tissue. The single-lobe steering range for OPA Type I and Type IV measured in tissue with >4 dB peak-to-background threshold are ~68 μm and 31 μm, respectively. At blue wavelengths, the beam width (FWHM) after propagating 50 μm was broadened by 18.7 μm to an average of 33.1 μm. At amber wavelengths, the mean FWHM of the emitted beams from OPA IV was 18.6 μm, which was 8.7 μm more than in non-scattering media. The broadening of the beam within the tissue reduced the number of achievable resolvable points from 4–5 to approximately 2 (single-lobe steering range/FWHM beam width measured in tissues) at 50 μm. Also, the beam width for OPA Type I broadens up to a similar scale as the single-lobe steering range (~67 μm) at 100 μm propagation distance, leading to no resolvable points beyond 100 μm.

### *In vivo* experiments

To validate the photostimulation in vivo, we implanted a hybrid probe with 4 OPA Type I emitters and 10 microelectrodes, as shown in Fig. 5a, in the cortex of Thy1-ChR2-YFP mice. Our beam profile analysis showed that OPA I achieved a single-lobe steering range of 68 μm after propagating a distance of 50 μm, when tuned over a ≈24 nm wavelength span. We targeted probe insertion into Layer V of the motor cortex as the brain region has a high level of expression of the ChR2[1]. A diode laser with an external cavity provided the tunable wavelength input to the probe from 484.3 to 491 nm with a tuning speed of 1–2 s over the full tuning range.

The laser scanning system used to address the OPA emitters could turn on one OPA at a time, with a switching speed of 5–10 ms. This speed is adequate for some applications of circuit mapping[1,10,41]. However, to permit neural stimulation at the temporal resolution of a single action potential, switching speeds should be below ≈1 ms[40]. Future optimizations could include the design of an integrated thermo-optic switch network on the neural probe, allowing OPA switching at ≈50 kHz and enabling concurrent use of multiple OPA emitters[22].

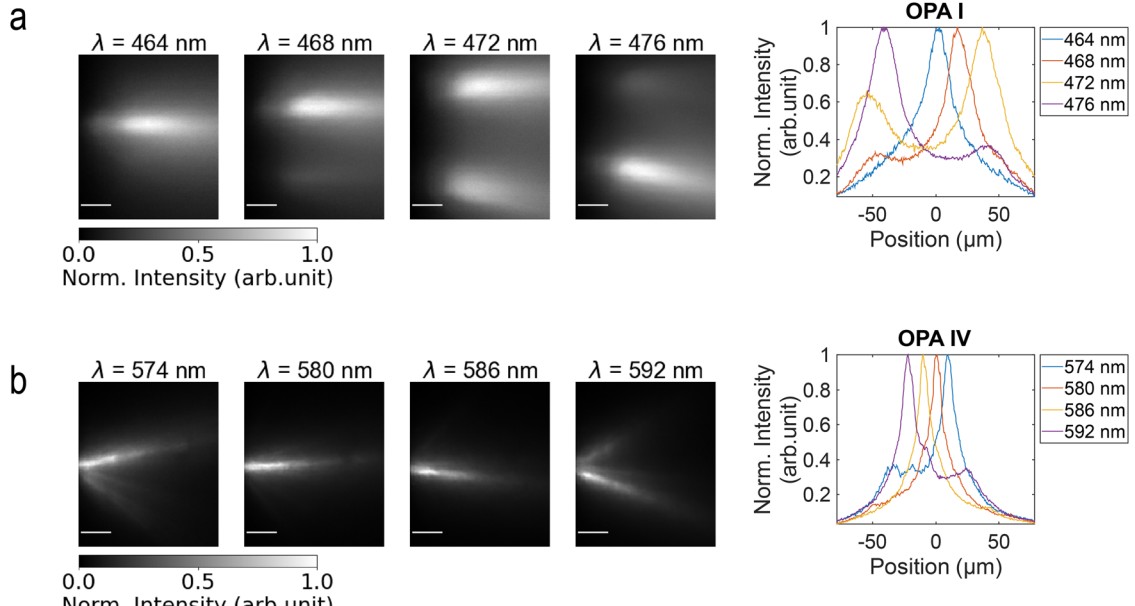

**Fig. 4 | Beam profile characterization in fixed brain tissue.** Measured intensity top-down beam profiles of **a** OPA I and **b** OPA IV in fixed tissue. The scale bars in (**a**) and (**b**) are 25 μm. On the right are radial line profiles of the measured images at a propagation distance of 50 μm showing the intensity versus lateral position of emitted beams as they are steered in tissue.

A snapshot of the recording trace before and during a photostimulation pulse train from the OPA nearest to the probe tip across 9 electrodes is presented in Fig. 5b (one electrode was not electrically connected). A 4 Hz pulse train with a 50-ms pulse width was applied for 2.5 s, repeated five times with a 10 s interval for recovery, resulting in a total of fifty 50-ms stimulation pulses. Subsequent spike sorting of the recording verified that the stimulated unit exhibited a clean spike waveform and minimal refractory period violation in its autocorrelogram (see Fig. 5c).

Increasing the optical output power from the OPA emitter resulted in a higher average stimulated firing rate and a reduced spike latency of the first spike after the onset of the optical pulse in the stimulated unit, as presented in Fig. 5d. Across the output power range of 0.17–1.22 μW (0.35–2.54 mW/mm² estimated with the $1/e^2$ beam size measured in fluorescein data), the average firing rate increased from 14 to 50 spikes/s, while the average spike latency shortened from 24 ms to 14 ms. These output powers are expected to be within a safe thermal operating range in tissue, as the input light to an implanted OPA device is estimated to be <20 μW, which is lower than the highest safe operating power dissipated by the μ-LED neural probe reported in ref. 18. The raster plot in Fig. 5e demonstrates a consistent stimulation response over all 50 pulse repetitions, with a 94% probability of detecting at least one spike within the optical pulse.

We have attempted to detect a difference in photostimulated spiking patterns as we steered the beam. However, the small steering angle constrained by the laser wavelength tuning range available at the in vivo experimental site yielded no discernible differences in the firing patterns across the electrodes. The laser wavelength tuning range of 485.5–489 nm resulted in a beam translation of ~9 μm at 50 μm propagation distance, which is less than the FWHM beam width measured in fluorescein solution (14.4 μm), leading to no resolvable pattern even under non-scattering conditions. The broadening of the beam in the tissue observed in Fig. 4a further reduced the spatial resolution of the beam.

Also, since the star couplers were not coated with optically opaque epoxy to prevent an increase in the size of the device in the animal experiments, the light that scattered from the star coupler (positioned at ≈140 μm from electrode E9) could have contributed to the observed photostimulated response in the experiment and broadened the stimulation volume. Despite these limitations, the experiment demonstrated that a neural probe with an array of OPAs and electrodes can perform simultaneous photostimulation and electrophysiological recording in vivo.

## Discussion and conclusion

In this study, we conducted a comprehensive characterization of four sidelobe-free OPA designs for optogenetic stimulation applications, with three designs using different grating curvatures (concave, straight, and convex) for blue wavelengths and one straight slab grating for amber wavelengths. The FPR for OPA Type I and III introduces an additional lateral translation of the beam at the grating emission site, increasing the steering range and the resolvable points close to the probe within the electrode detection range (<140 μm). In contrast, OPA Type II with a straight slab grating and no FPR slab compromises the steering performance near the probe for a more compact footprint. Supplementary Note 2 demonstrates that further reducing the device footprint is feasible by using higher index SiN material and thicker waveguides, enabling the integration of 8 OPA emitters on a 70 μm wide probe shank.

We also confirmed through in vivo experiments using a neural probe with OPA Type I that the OPAs emitted sufficient power for optogenetic stimulation in Thy1-ChR2 mice, with neuronal spiking activity recorded by the probe's electrodes. However, the limited wavelength tuning range of the laser at the experimental site restricted our ability to leverage the OPAs' full steering capabilities and validate their potential to perform selective stimulation. Despite this limitation, based on ChR2 sensitivity to optical intensities[44] and using the FWHM beam width measured in tissue (see Fig. 4a) to evaluate resolvability, we estimate that by optimizing the beam power to induce a spiking probability of ~65% for neurons at the center of the beam, the neurons outside of the beam width would be activated with a probability of <35%. Therefore, the current OPA design could expect to induce variations in neural firing patterns by scanning the beam (FWHM ~33 μm) to stimulate neurons positioned ~50 μm from the probe at the two maximum emission angles (lateral distance of ~68 μm) and photostimulating the targeted neurons at submaximal response levels.

The OPA emitters can steer beams over a continuous area, achieving multiple resolvable points using just one input waveguide per device. These devices can have a scalability benefit over probes using multiple discrete grating emitters for patterned illumination, with each grating requiring its own waveguide input. Supplementary Note 3 shows that with OPA Type III

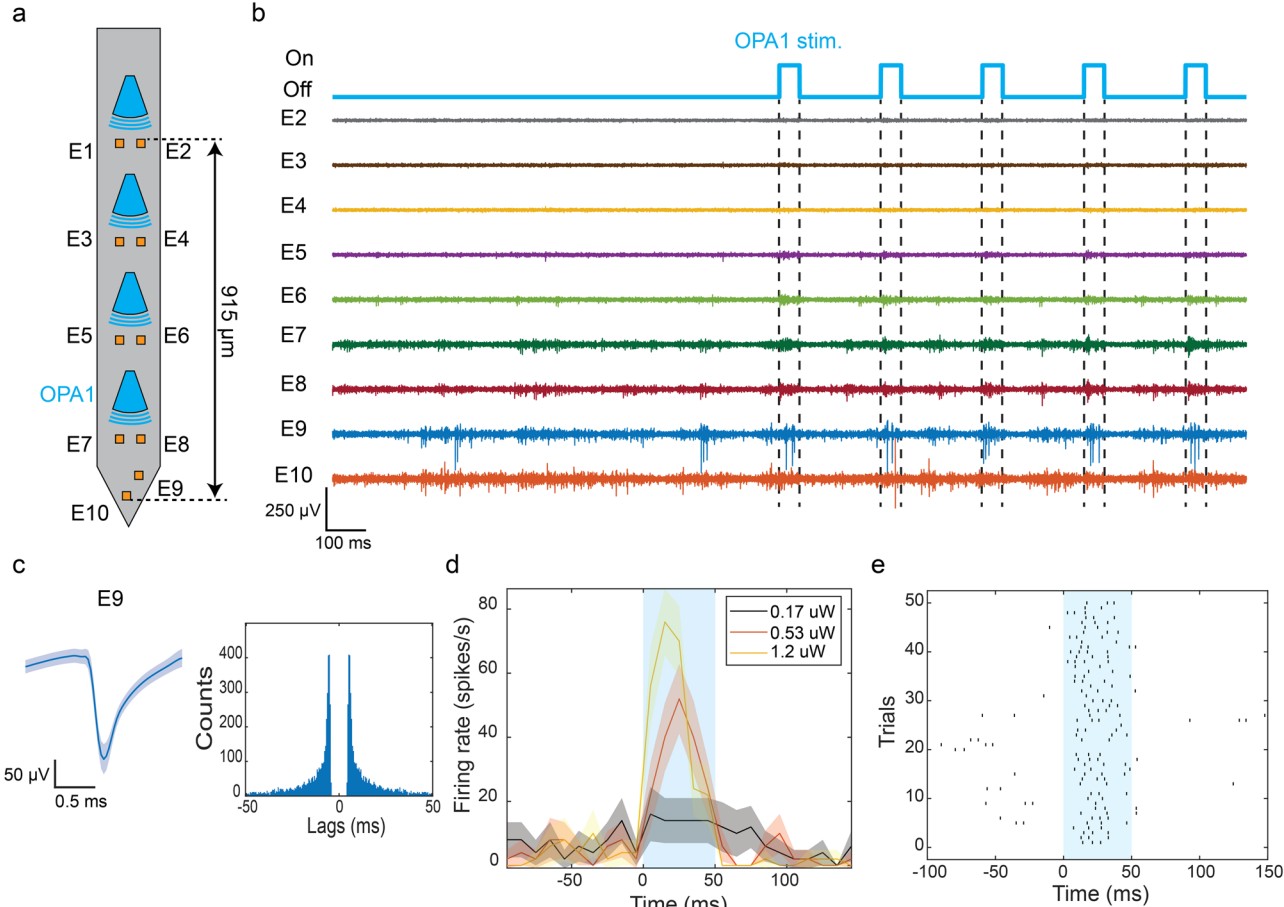

**Fig. 5 | Demonstration of optogenetic stimulation with the OPA Type I emitter in in vivo awake head-fixed mice. a** Layout of the OPA probe used in the in vivo experiment. It contained 4 OPA Type I emitters and 10 microelectrodes. **b** Snapshot of the filtered electrophysiological recording (common average referencing (CAR) + bandpass filtered (300–6000 Hz) + artifact subtraction). OPA1 labeled in (**a**) was used for optogenetic stimulation. The trigger signals of the optical pulses are indicated above, with dashed lines marking the onset and offset of the pulse. Electrode indices are labeled on the side. The output power was 1.22 μW. **c** Waveform and autocorrelogram of a sorted unit recorded on electrode E9. The solid blue line represents the mean waveform and the shaded area represents the standard deviation of the waveform. **d** The mean peristimulus time histogram ($n = 50$ trials) of the unit in (**c**) under varying optical stimulation power from OPA1. The solid lines represent the mean firing rate and the shadings represent the standard error. **e** Raster plot of the unit recorded with electrode E9 during optical stimulation from OPA1 in 50 trials, each consisting of 50-ms optical pulse stimulation at a power level of 1.22 μW. The blue shaded areas in (**d**) and (**e**) indicate the stimulation period.

as an example, using OPA emitters can realize more resolvable points in non-scattering media than discrete grating approaches on a probe with a shank width larger than 110 μm. However, the number of points that can be resolved is currently limited to ~2 in tissue due to beam broadening from scattering. Strategies for improving steering range and reducing beam size discussed next could enhance the OPAs' performance in scattering media. Moreover, while the OPA probes have a lower number of addressable emitters compared to the highly integrated μ-LED or μ-OLED probes demonstrated in refs. 18,19, the beams emitted from the OPAs can achieve a narrower divergence (<±3 degrees at half maximum beam intensity for OPA Type I vs. > ±35 degrees for optoelectronic emitters with Lambertian beam profiles[19]). This property of the OPA emitters could offer better selectivity in targeting neurons within the electrode detection range.

The proof-of-concept experiments highlight the need for continued improvements to the OPA designs and experimental methods to achieve spatially selective optogenetic stimulation. The main challenge for adopting the proposed OPAs in targeted optogenetic stimulation is the limited number of resolvable points achieved in tissue. Several near-term strategies can be implemented in the OPA designs to increase the number of achievable resolvable spots. First, future experiments should leverage the reduced optical scattering of longer wavelengths by conducting experiments with OPAs designed for red wavelengths in combination with red-shifted opsins (such as ChRmine at 635 nm[47]) to reduce beam broadening in tissue.

Switching the operating wavelength from 470 nm to 635 nm can extend the scattering length by 30−40%[48,49]. Second, the steering range of the OPAs can be further increased while maintaining beam divergence by (1) reducing the phased array pitch and (2) increasing the number of emitters in the phased array to preserve the aperture size. For instance, OPA Type IV is expected to gain a 1.8-fold increase in steering range by reducing the phased array pitch from 500 nm to 300 nm based on the OPA steering range equation in ref. 50. As the number of emitters increases, thicker waveguides or higher index waveguide materials can be used to minimize the increase in OPA footprint, as discussed in Supplementary Note 2. Finally, future OPA designs could adopt out-of-plane focusing grating strategies demonstrated in refs. 28,51 on the slab grating or introduce a parabolic phase shift at the delay lines[52] to generate steerable focused beams, thereby reducing beam broadening in tissue and localizing the stimulation near the focal point where the intensity is the highest. Reference 28 has demonstrated that it is possible to achieve focused beams with widths <10 μm located 50 μm from the probe in tissue with grating-based emitters. A long-term solution could involve implementing active thermal phase shifters for each waveguide emitter in the end-fire phased array. This approach would (1) provide greater control on beam shaping beyond beam steering along an arc (i.e., adjusting beam focusing depth[22]) and (2) enable phase optimization to compensate for scattering-induced phase distortions[53]. The firing rate of the targeted/untargeted neurons could be explored as the feedback signal to optimize the beam selectivity

on stimulating neurons, as the firing rate is dependent on the incident beam intensity as seen in Fig. 5d. However, to realize high-density integration of OPA emitters with wavefront shaping capabilities, the development of energy-efficient, compact phase shifters that can be placed on the shank and implanted in tissue without dissipating excessive heat is necessary.

Alongside these device enhancements, our experimental methods should also be refined to conclusively show spatially selective optogenetic stimulation in vivo. Using a tunable laser source matching with the FSR of the OPA is essential for future in vivo experiments to investigate the stimulation effect across the full steering range of the device. Moreover, applying light-absorbing epoxy will help block stray and scattered light, improving the confinement of the effective photostimulation volume. On the chip, scattered light from the photonic circuit components, such as the star coupler, can be blocked using the top metal routing layers. On the biological side, using soma-targeted opsins would spatially restrict opsin expression within the cell body[47,54], minimizing undesired stimulation of axons and dendrites, which could otherwise broaden the stimulation resolution[9]. Additionally, administering low concentrations of glycerol through drinking water to rodents[55] could be adapted for the optical clearing of living brains, thereby reducing tissue scattering. Lastly, switching activity monitoring modalities from electrophysiological recordings to all-optical brain interrogation, where neural activity is recorded by calcium or voltage imaging[56,57], can provide direct observation of the relative positions between the OPA beam and the stimulated neurons at cellular resolution, enabling a more precise measurement of the effective stimulation area of the beam at different emission angles.

In addition to future work aimed at enhancing experimental and device strategies, we believe that it is also crucial to further study how different factors influence the beam profiles measured in tissue. This will aid in understanding the differences in beam profiles observed between the experimental measurements in the mouse cortex and the estimates from the scattering simulation model reported in ref. 58 (see Supplementary Note 4). In the simulation model, tissue scattering was simulated by propagating the beam through a series of randomized phase masks with controlled statistical parameters. The discrepancies observed between the simulation and the experimental results may be due to several factors, including (1) higher optical scattering in PFA-fixed tissue samples compared to the fresh samples used in simulations as reported in refs. 59–61, (2) uneven tissue conformity to the probe surface causing additional scattering and beam distortion, and (3) the absence of an experimentally validated method to generate phase masks for tissue scattering simulations. Future beam profile measurements using fresh brain slices could mitigate the first two factors, as they would better reflect the scattering properties in in vivo environments and ensure better conformity to the probe surface due to reduced tissue stiffness[62]. Additionally, employing quantitative phase imaging techniques[63] to extract spatial index variations in thin brain slices could guide the development of more realistic phase masks for simulating tissue scattering, ultimately improving the accuracy of the simulation model.

Finally, to ensure the OPA neural probes are compatible with chronic animal experiments, we also need to improve the durability of the fiber-attached neural probes. The lifespan of these probes largely depends on the stability of the output power. We observed a power drop of 1–3 dB after each in vivo experiment that lasted for 5–6 h. This power variation can limit the lifetime of the probe, specifically when the maximum output power falls below the intensity threshold (0.1–1 mW/mm[21]) required for optogenetic stimulation. We suspect that this gradual decrease in power over time is due to misalignment between the fiber cores and edge couplers, which likely resulted from two specific factors. First, the UV epoxy may exhibit insufficient bonding strength to maintain micron-scale alignment between the fiber cores and edge couplers. Second, there could be a slow epoxy shrinkage upon exposure to input blue light, as our previous studies[35] have reported similar but greater power drops when operating the probe at near-UV wavelengths (405 nm), closer to the photoinitiation wavelength of the epoxy. Studies focused on identifying a more reliable UV-curing epoxy are ongoing to improve the reusability of the probe.

In summary, we have reported passive single-lobe steering OPAs for blue and amber wavelengths on Si neural probes for in vivo optogenetic applications. A unique photonic circuit architecture that combines an in-plane phased array with a light-emitting grating-enabled single-lobe beam steering in a compact footprint without meeting the half-wavelength pitch criterion. While the beam-steering performance of the OPAs in tissue remains an area that needs further improvement, the design principles, device performance data, and limitations observed in tissue experiments serve as a foundation for future development of OPA emitters on neural probes for precise targeting optogenetic stimulation in deep brain regions, enabling a new class of neural circuit activity mapping experiments.

## Methods
### Device fabrication
The neural probes with OPAs were fabricated by Advanced Micro Foundry (AMF) on 200-mm diameter Si wafers. Initially, $SiO_2$ and SiN were deposited on the Si wafer for the waveguide cladding and core. The choice between LPCVD or PECVD for SiN depended on the required refractive index for specific OPA designs. The thickness and type of SiN for each OPA design are described in Table 1. Subsequently, 193-nm Deep Ultraviolet (DUV) photolithography and reactive ion etching were applied to the SiN layer to pattern the photonic circuits. Then, $SiO_2$ was deposited above the SiN to encapsulate the waveguide core within the $SiO_2$ cladding. Three layers of aluminum (Al) routing and TiN surface electrodes were deposited above the waveguide structure for electrophysiological recording electrodes and metal wiring. The outline of the probe was defined using deep trench etching from the front of the wafer, and the probes were released from the wafer by backgrinding the wafer to a thickness of about 100 μm. An additional laser roughening treatment, developed and reported in ref. 26, was performed on the electrodes to increase their effective surface area. This step reduces the electrode impedance to below 2 MΩ, an impedance threshold to achieve an acceptable signal-to-noise ratio (SNR) for electrophysiological recording[64]. In this treatment, a femtosecond laser (Monaco 1035, Coherent) was used as it has been shown to be effective at creating nanostructures on titanium-based implants[65]. A two-photon microscope (Ultima 2Pplus, Bruker, Billerica, MA, USA) was used to control the beam to scan across the electrode surface. Then, the electrodes were visually inspected under an optical microscope to confirm that they were intact and their surface color had darkened.

### Neural probe packaging
The neural probes were packaged on different substrates depending on their intended use. The packaging procedure was similar to the methods described in refs. 26,35. For beam profile measurements, the probes were bonded to metal holders using heat-curable metal epoxy (Ablebond 84-1LMIT1, Loctite, Stamford, CT, USA). Prior to attaching the multicore fiber (MCF) to the chip, a small amount of optically opaque epoxy (EPO-TEK-320, Epoxy Technology, Billerica, MA, USA) was applied to the star coupler on the OPA emitters to block stray light from the device. A custom 10- or 16-core MCF fiber was then actively aligned to the probe edge couplers using a 5-axis fiber alignment stage with a fiber rotator for rotational alignment. After achieving optimal alignment between the MCF and the chip, we incrementally applied small amounts of low-shrinkage UV-curable epoxies (OP-67-LS and OP 4-20632, Dymax Co., Torrington, CT, USA) to the probe and MFC, followed by UV curing with a UV LED system (CS2010, Tholabs, Newton, NJ, USA) after each drop to minimize alignment drift due to epoxy shrinkage. After the MCF was secured with epoxy, an additional 5 min of epoxy was applied to the back of the probe holder and MCF to provide stress relief. Finally, optically opaque epoxy was applied to cover the probe base and fiber to prevent stray light from reaching the samples.

For the probe used in the in vivo animal experiment, we mounted the probe on a custom printed circuit board (PCB) with a metal block added to the bottom of the probe to address clearance issues between the MCF and the base of the PCB. The probe was bonded to the metal block and PCB with the heat-curable metal epoxy (Ablebond 84-1LMIT1, Henkel, Germany).

The probe was then wire bonded to the PCB with Au wires. To minimize light-induced artifacts in electrophysiological recordings, a UV-curable encapsulating epoxy (Katiobond GE680, Delo, Germany) and optically opaque epoxy were applied to minimize stray light from the fiber reaching the wire bonds. Similar MCF-to-chip attachment procedures described above were then performed after electrical packaging, as optical packaging requires higher alignment accuracy. We did not use the optically opaque epoxy to cover the star coupler for the probe used for in vivo experiments, as it increases the size of the implant. Finally, the probes were left undisturbed for at least 12 h to allow the epoxy to fully cure before removal from the packaging assembly.

## Optical scanning system and tunable laser sources

We used a custom-built optical scanning system, similar to the one described in refs. 26,35,38, to address the emitters on the probe with light from an external laser source. This system used a micro-electromechanical system (MEMS) mirror and custom optical lenses to direct and focus the input light onto each fiber core of the MCF (core size of ~2.6 μm). The insertion loss between the input of the scanning system and the MCF output was approximately 2–3 dB. We used a program scripted in MATLAB to search for the MEMS coordinates corresponding to the fiber cores on the MCF and to control the MEMS during the experiments. The scanning system used a fiber input port for adaptability to various laser sources. For beam profile characterization in fluorescent solutions and mouse brain slices, a supercontinuum white laser (SuperK Fianium, NKT Photonics) with a tunable filter (LLTF Contrast, NKT Photonics), that has a bandwidth of 1–2 nm and covers a range of 400–1100 nm, was employed as the input light source. For in vivo experiments, an external cavity laser with motorized wavelength control was used to tune the wavelength with an estimated linewidth of 100 kHz, between 484.3 and 491 nm (TOPTICA Photonics Inc., DLC DL pro tunable laser system with integrated fiber coupler). We constructed a free-space-to-fiber-coupling stage for each laser source to couple light into a single-mode fiber (460-HP, Nufern Inc.), which was then connected to the scanning system. An optical shutter and a variable neutral density filter were added in between the laser and the fiber-coupling stage used in the in vivo experiment for gating the input beam and controlling the optical power.

## Animals

The measurements in fixed brain slices were carried out at the Max Planck Institute of Microstructure Physics in Halle, Germany. The in vivo experiments were conducted at the Krembil Brain Institute in Toronto, Canada. Fixed brain slices were provided by the Fraunhofer Institute for Cell Therapy and Immunology and the surgical procedures to extract the brain were performed in accordance with German laws. The in vivo experimental procedures described here were reviewed and approved by the animal care committees of the University Health Network in accordance with the Canadian Council on Animal Care guidelines. Adult male and female Thy1-ChR2-YFP mice (The Jackson Laboratory, Bar Harbor, Maine, stock number 007612) were kept in a vivarium maintained at 22 °C with a 12-h light on/off cycle. Food and water were available ad libitum.

## Preparation of fixed mouse brain slices stained with fluorescent dye

To assess the beam profile emitted from each OPA type within a scattering medium, we captured beam propagation in fixed brain tissue slices. We used the whole brain extracted from wild-type mice (C57BL) at the age of 30–150 postnatal days. The whole brain samples were immersed in 1.5–2% paraformaldehyde (PFA) for 8–12 h of fixation. Then, they were stored in 1× phosphate-buffered saline (PBS) at 4 °C before use. Before experiments, the whole brains were sectioned into 2-mm-thick coronal slices using a brain matrix (Alto Brain Matrix stainless steel 1 mm mouse coronal 45–75 g; Harvard Apparatus) and stirrup-shaped blades (Type 102, Carl Roth GmbH + Co. KG, Karlsruhe, Germany). Following the staining strategies reported in ref. 66, the slices were then permeabilized with 0.3% Triton X solution for 30 min, followed by three 5-min washes in PBS. The slices were then immersed overnight in 100 μM fluorescein or Texas Red fluorescent dye, depending on the operating wavelength of the OPA being tested (450–484 nm: fluorescein, 574–598 nm: Texas Red). The slices were ready for use the next day after three additional 5-min washes in PBS.

## Beam profile characterization

We evaluated the steering range and beam profile of each OPA type in non-scattering and scattering media. The input light was set to TM-polarized light for maximum power output from the OPA grating using an in-line fiber polarization controller. For the non-scattering medium, the shank of the probe was immersed in either 100 μM fluorescein or Texas Red fluorescent dye, depending on the operating wavelength for the OPA type (450–484 nm: fluorescein, 574–598 nm: Texas Red). A 4-axis micro-manipulator (uMp-4, Sensapex, Oulu, Finland) precisely controlled the probe's movement while top-down beam profiles were captured using a wide-field epifluorescence microscope (Cerna, Thorlabs) equipped with a 10× objective (M Plan Apo 10×, NA = 0.28, Mitutoyo Deutschland GmbH), appropriate filter cubes (49002, Chroma Technology Corporation, Bellows Falls, VT, USA for GFP and mCherry-C-000-ZERO, Semrock, Rochester, NY, USA for mCherry), and an sCMOS camera (Prime BSI, Teledyne, Photometrics, Tucson, AZ, USA). Additionally, side-view beam profiles were acquired with a microscope to examine the beam profile thickness and potential stray light emission from the probe other than the OPA grating. The side-view microscope consisted of a variable magnification microscope (12× Zoom Lens System, Navitar, Rochester, NY, USA) equipped with a 5× objective, a GFP emission filter (MF525-39, Thorlabs) or a Texas Red emission filter (MF630-69, Thorlabs), and a CMOS camera (Grasshopper3, USB 3.0, Teledyne FLIR, Wilsonville, OR, USA). A supercontinuum white laser (NKT SuperK Fianium) with a tunable filter with a tuning range of 400–1100 nm provided the input light source to the probe.

To evaluate beam profiles in scattering media, the probe shank was inserted into fixed brain tissue prepared following the procedures outlined in the "Methods" section. The same fluorescence microscope system described above was utilized to capture top-down beam profiles within the scattering medium. For capturing the beam profile close to the tissue surface, the OPA emitter was first inserted fully into the tissue, and a series of beam profiles over the complete FSR of the OPA was acquired. This image acquisition process was repeated after retracting the probe at 30–50 μm increments until the OPA emitter was out of the tissue.

## Experimental procedures for awake head-fixed animal experiments

The head-fixed animal experimental protocol was similar to the process reported in our previous work presented in ref. 26. In preparation for the awake head-fixed experiment, the headplate was mounted onto the mouse skull 2–4 days prior to the experiment. Thy1-ChR2-YFP mice at the age of 60–90 postnatal days were used. The mouse was first anesthetized with 5% isoflurane/oxygen by induction and maintained with 1–2% isoflurane/oxygen. It was then secured in a stereotaxic frame using ear bars (Model 902, David Kopf Instrument, Tujunga, California). Before mounting the headplate, two bone screws (Item No. 19010-10, Fine Science Tools) were implanted for ground and reference electrodes. We followed the same ground and reference configuration described in ref. 67. The insertion positions (stereotaxic coordinates of AP: 0 to −0.5 mm, ML: 1.2 mm for targeting motor and somatosensory cortex, with the origin aligned to bregma) were also marked using a dental drill. Finally, the headplate was attached to the mouse skull using dental cement (C&B Metabond, Parkell, Edgewood, NY, USA). The mouse was placed in a separate cage for recovery.

On the day of the experiment, the mouse with a headplate was anesthetized following the same protocol as before. We then used a dental drill to open a circular craniotomy of 1–2 mm diameter at the previously marked position for probe insertions. The dura was left intact to minimize the

motion artifact caused by brain pulsation. We then placed the mouse body in a 3D-printed cone under the microscope to restrain mouse movement, and the headplate was attached to the head bar posts. The probe was mounted on a micromanipulator for precise translation and insertion speed control. The reference and ground screws were connected to the corresponding pins on the Intan headstage, and we attached an additional ground wire from the ground plane of the optical table to strengthen the ground connection.

To ensure successful dura penetration, the probe was inserted into the brain at a speed of 0.5–1 mm/s for the first 500–900 μm. Following confirmation of the probe insertion, the probe was retracted by 200–500 μm to allow the brain to relax and was held in this position for 10–15 min. Subsequently, the probe was advanced at a slower speed of 1–2 μm/s to minimize tissue damage until a depth of 1–1.4 mm was reached based on the micromanipulator-measured insertion depth, where the electrodes near the tip of the shank were positioned in cortical layers V and VI. The probe was rested for approximately 30 min before the start of the photostimulation.

Prior to photostimulation, the input wavelength was set to 485.5 or 490 nm, and an electrically driven polarization controller at the laser output was used to set the input light to TM polarization based on the scattered intensity from the probe captured by the microscope camera. For optogenetic stimulation, a 4 Hz pulse train with 50 ms optical pulses was applied for 2.5 s, repeated five times with a 10 s recovery period between each pulse train.

After the experiment, the probe shank was immersed in 1% Tergazyme solution (Sigma-Aldrich, St. Louis, MO, USA) for 2 h, followed by a 10–20 min wash in Milli-Q water. The probe was air-dried for subsequent experiments.

## Electrophysiological data analysis

The raw electrophysiological data recorded from the Open Ephys acquisition board[68] were preprocessed in Python before spike detection. The spike analysis steps were identical to the process reported in ref. 26. First, we applied common average referencing using electrode channels that detected fewer spikes as the reference signal. The signal was then filtered with a third-order Butterworth bandpass filter (300–6000 Hz). To reduce the amplitude of light-induced artifacts, we calculated the average light-induced artifact waveform for each OPA at the same stimulation power and subtracted it from the onset and offset of the light pulse. Additionally, a blanking window of 1 ms preceding and 2.5 ms following the start and end of the optical pulse was applied to minimize the artifact's impact on spike detection accuracy.

For spike detection and clustering, we utilized the Spyking Circus package[69], followed by manual curation of spike clusters via the phy GUI interface[70]. To address double counting of the same spikes during the manual merging process, we removed one of the paired spikes that have an interspike interval of less than 0.5 ms. The final selected spike clusters follow four criteria: (1) isolation distance >10, (2) likelihood ratio ≤0.3[21,71], (3) SNR ≥3[72], and (4) the percentage of spikes with refractory period violation (<2 ms) <2%[73].

## Reporting summary

Further information on research design is available in the Nature Portfolio Reporting Summary linked to this article.

## Data availability

Raw electrophysiological data, beam profile images, and the source data for the figures are available at: https://doi.org/10.17617/3.1X2HCV.

## Code availability

All codes used in this research are available from the corresponding authors upon reasonable request.

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

## Acknowledgements

The authors thank Holger Cynis, Ines Koska, and Stefanie Geißler from the Fraunhofer Institute for Cell Therapy and Immunology for providing mouse brain tissue samples and Andrei Stalmashonak and Hannes Wahn for their assistance in setting up the optical systems. Additionally, the authors would like to thank Hanne Wahn for the discussions on the scattering simulations. The authors also gratefully acknowledge funding support from the Max Planck Society, Natural Sciences and Engineering Research Council of Canada, and the Canadian Institute of Health Research.

## Author contributions

A.S., Y.J., and J.C.C.M. conceived the design approach. F.D.C., A.S., and Y.J. performed the device simulations, designed the probes with inputs from W.D.S. and J.K.S.P., and A.S., T.X., and A.G. laid out the designs. H.C., X.L., and P.G.Q.L. were responsible for the wafer fabrication. Devices were packaged by F.D.C., A.S., and T.X. A.S., F.D.C., and W.D.S. characterized the devices. M.G.K.B. prepared the fixed tissue samples. F.D.C., A.S., H.M.C., and M.M. conducted the animal experiments. F.D.C., A.S., and J.K.S.P. analyzed the data. A.S., F.D.C., and J.K.S.P. co-wrote the manuscript with inputs from other co-authors. The project was completed under the supervision of T.A.V., W.D.S., and J.K.S.P.

## Funding

## Competing interests

The authors declare no competing interests.
