## [Transparent Peer Review file · Communications Engineering]

This manuscript has been previously reviewed at another journal. This document only contains information relating to versions considered at *Communications Engineering*.

Implantable silicon neural probes with nanophotonic phased arrays for single-lobe beam steering

Corresponding Author: Ms Ankita Sharma

Version 0:

Reviewer comments:

Reviewer #1

(Remarks to the Author)

This paper presents an on chip integrated photonics platform for optical beam steering within tissue.

The authors have addressed most of the comments on the previous version satisfactorily. There are two remaining points to clarify and address:

1-The authors claim that the main reason they did not observe beam steering in their experiments in scattering media is the limited steering range. This claim is not substantiated. While this is a limitation of their implementation, the authors do not provide any clear reasoning or evidence that if this issue is fixed, they can resolve targets in real tissue. I would argue that after a few scattering events (e.g., one transport mean free path), light is diffused almost completely and increasing the steering range may not significantly help. They authors need to clarify this point as it is a core claim of the manuscript.

2- The question/suggestion about adaptive optics was more to help the authors think about the concept rather than the implementation. It is appreciated that the authors discuss thermal tuning as a mechanism for implementing a programmable beam steerer that can be used in this context. However, the more important question is how adaptive optics can be implemented where getting feedback from light propagation in scattering tissue is not easily possible. It would be helpful if the authors discuss (a) how the beam broadening due to scattering in tissue affects the performance of their system and (b) how the issue of scattering, especially in non-homogeneous media can be dealt with.

Reviewer #2

(Remarks to the Author)

Referee Report on the Revised Manuscript:

This paper introduces a neural probe design with a grating-based light emitter and an integrated optical phased array for steerable optogenetic stimulation. It demonstrates beam steering in transparent media but falls short in scattering media. I have reviewed the revised version of the manuscript, and I am pleased to note that the authors have satisfactorily addressed the comments raised in the initial review. The revisions have significantly improved the clarity and overall quality of the manuscript, and I believe that the work can be recommended for publication in Nature Comm. Engg. I appreciate the additional discussion provided in the supplementary section, particularly the section regarding scaling constraints and design trade-offs. This discussion adds significant value to the work, making it much more comprehensive and thorough. The insights on how these constraints affect design decisions are particularly useful for the audience seeking design guidelines for photonic devices for biomedical applications.

I have a few minor but important comments that should be addressed to further strengthen the manuscript.

1. It would be useful to provide citations/sources for the range of refractive index considered for PECVD and LPCVD SiN (Lines #106 - 109 in the manuscript)

2. In the in vivo experiment section, it is stated: "The laser scanning system used to address the OPA emitters, could turn on one OPA at a time, with a switching speed of about 5-10 ms."

I would appreciate it if the authors could provide more detail on whether this switching speed is sufficient for typical optogenetic experiments. If not, it would be helpful to suggest hardware or system-level designs that could enable faster

switching or even concurrent use of multiple OPAs for stimulation

3. In the “Beam Profile Characterization” section, the authors mention: “The tunable laser source has a response time of 55 ms with a 1 nm step size, resulting in a tuning speed of approximately 1.1 seconds across the full FSR.” Could the authors comment on whether this tuning speed is sufficient for the intended/envisioned experiments using the neural probe? If not, what optimizations or modifications might be considered to improve the speed to meet the experimental demands?

Overall, the manuscript presents notable advancements in the field, and with these minor revisions, I believe it will be even more impactful.

Version 1:

Reviewer comments:

Reviewer #1

(Remarks to the Author)

Thanks for addressing the comments and clarifying the points in the manuscript. The authors have addressed my comments in a satisfactory way.

Reviewer #2

(Remarks to the Author)

The manuscript presents an integrated photonic phased array for in vivo beam steering, enabling spatially controllable optogenetic stimulation. The authors have satisfactorily addressed all the minor comments raised in my previous review. The added clarifications, along with the inclusion of relevant references, have improved the manuscript's clarity and contextual understanding of the work.

At this point, I do not have any further comments or questions for the authors, and in my opinion, the manuscript can be considered for publication.

Dear Reviewers,

We'd like to thank you for your insightful comments. Our responses to the comments are below in blue with the relevant changes in the manuscript highlighted in yellow. Some light additional edits to the work are mentioned at the end of the response. Also, for quicker access, references to publications are provided using doi links in this letter. We hope these additional responses address your comments and the manuscript would be recommended for publication in *Communications Engineering*.

Sincerely,
Fu-Der Chen and Ankita Sharma

Reviewer #1 (Remarks to the Author):

This paper presents an on chip integrated photonics platform for optical beam steering within tissue.

The authors have addressed most of the comments on the previous version satisfactorily. There are two remaining points to clarify and address:

We thank the reviewer for providing the comments to help improve the work.

1-The authors claim that the main reason they did not observe beam steering in their experiments in scattering media is the limited steering range. This claim is not substantiated. While this is a limitation of their implementation, the authors do not provide any clear reasoning or evidence that if this issue is fixed, they can resolve targets in real tissue. I would argue that after a few scattering events (e.g., one transport mean free path), light is diffused almost completely and increasing the steering range may not significantly help. The authors need to clarify this point as it is a core claim of the manuscript.

We appreciate the reviewer's comment and understand that this was not clearly explained in the manuscript. Regarding our comment on the *in vivo* experiment having limited steering range, we were referring to performing selective stimulation on neurons proximal to the probe (50 \$\mu\text{m}\$, about one scattering length in blue wavelengths Ref. S4 S5). In the Discussion, we mentioned that if we had access to the full steering range with the current OPA I design, "we estimate that by optimizing the beam power to induce a spiking probability of \$\sim 65\%\$ for neurons at the center of the beam, the neurons outside of the beam width would be activated with a probability of \$< 35\%\$." This estimation was based on the ChR2 sensitivity to optical intensities curve provided in Fig.6 A of Ref.44 and using the average FWHM beam width at 50 \$\mu\text{m}\$ propagation distance extracted from our experimental beam profile data in tissue (Fig. 4A, FWHM of 33 \$\mu\text{m}\$ with single lobe steering range of 68 \$\mu\text{m}\$ ). We also plotted the beam profiles at 50 \$\mu\text{m}\$ from the probe at two extreme single-lobe steering ranges in Figure R1 for comparison of their beam intensities. This analysis provides evidence that the current device (with full steering range) can induce some variations in firing patterns on neurons \$\sim 50\$ \$\mu\text{m}\$ from the probe, which corresponds to the electrode detection range for large amplitude spikes (\$> 60\$ \$\mu\text{V}\$ ) as discussed in Section C.

Figure R1. Radial line profiles in tissue at a beam propagation distance of 50 μm for beams at two extreme single lobe steering angles emitted from OPA Type I. The profiles are extracted from Figure 4a in the manuscript.

Selective stimulation at hundreds of micrometers in depth is challenging with one-photon excitation due to tissue scattering. However, this may not be necessary for simultaneous optogenetic stimulation and electrophysiological recording, as the electrodes cannot reliably detect spikes from neurons beyond 140 μm [Ref. 43]. The closer the neurons are to the probe, the easier it is to sort the spikes into well-isolated units.

We have added the assumption that the neuron should be around 50 μm away from the probe to observe the selective stimulation effect:

Despite this limitation, based on ChR2 sensitivity to optical intensities [Jackman2014] and using the FWHM beam width measured in tissue (see Fig. \ref{fig:tissue}a) to evaluate resolvability, we estimate that by optimizing the beam power to induce a spiking probability of $\sim 65\%$ for neurons at the center of the beam, the neurons outside of the beam width would be activated with a probability of $< 35\%$. Therefore, the current OPA design could expect to induce variations in neural firing patterns by scanning the beam (FWHM $\sim 33 \text{ }\mu\text{m}$) to stimulate neurons positioned $\sim 50 \text{ }\mu\text{m}$ from the probe at the two maximum emission angles (lateral distance of $\sim 68 \text{ }\mu\text{m}$) and photostimulating the targeted neurons at submaximal response levels.

To further enhance the device performance and work toward independent stimulation of neurons proximal to the probe, we have proposed several future directions in our response to Q2, which include increasing the steering range and reducing the beam size to ultimately improve the beam contrast and the number of resolvable points in tissue.

2- The question/suggestion about adaptive optics was more to help the authors think about the concept rather than the implementation. It is appreciated that the authors discuss thermal tuning as a mechanism for implementing a programmable beam steerer that can be used in this context. However, the more important question is how adaptive

optics can be implemented where getting feedback from light propagation in scattering tissue is not easily possible. It would be helpful if the authors discuss (a) how the beam broadening due to scattering in tissue affects the performance of their system and (b) how the issue of scattering, especially in non-homogeneous media can be dealt with.

We thank the reviewer for the comments. Optical scattering in tissue degrades beam quality, which limits the stimulation selectivity. Our beam width measurements from OPA Type I shown in Figs. 3 and 4 illustrate this effect. At a 50 μm propagation distance, beam broadening observed in tissue reduces the number of achievable resolvable points from 4-5, as measured in fluorescein, to approximately 2. The FWHM beam width (66 μm) covers the single lobe steering range (67 μm) as the beam propagates to 100 μm in tissue, while the resolvable points measured in fluorescein remain unchanged. This observation shows the limitation of the current device to perform beam steering beyond 100 μm in tissue.

To improve beam selectivity in tissue near the probe (~50-100 μm), we have suggested several future directions to explore, including (1) implementing the OPA design in red wavelengths for red-shifted opsins to reduce optical scattering, (2) reducing the phased array pitch while maintaining the phased array aperture size to increase steering range and (3) designing OPA emitters with a steerable focused beam. To put the improvements in context, switching the operating wavelength from 470 nm to 635 nm could extend the scattering length by 30 - 40 % [Ref. 48,49]. For suggestion 2, reducing the phased array pitch for OPA Type IV as an example from 500 nm to 300 nm would theoretically increase the steering range by ~1.8 times, based on the OPA steering range of Equation 4 in Ref. 50. This enhancement would increase the resolvable points to ~ 3. Last, emitting a focused beam can help localize the stimulation near the focal point where the intensity is the highest. Work in Ref. 28 has shown it is possible to generate a focused spot at 50 μm away from the probe with FWHM beam width of <10 μm using grating-based emitters.

On the biological side, we can administer low concentrations of glycerol through drinking water to rodents as a method for clearing the tissue in the living brain for reduced tissue scattering [Ref. 49]. As this method is still in its early research phase, further experimentation is required to better assess the amount of reduction in beam width that can be realized with this method.

We have improved our discussion on scattering effect on beam resolution and the benefit each improvement brings in the following sentences:

C. Beam Profile Characterization:

The broadening of the beam within the tissue reduced the number of achievable resolvable points from 4-5 to approximately 2 (single-lobe steering range/FWHM beam width measured in tissues) at 50 μm . Also, the beam width for OPA Type I broadens up to a similar scale as the single-lobe steering range ($\sim 67\mu\text{m}$) at 100 μm propagation distance, leading to no resolvable points beyond 100 μm .

Discussion:

First, future experiments should leverage the reduced optical scattering of longer wavelengths by conducting experiments with OPAs designed for red wavelengths in combination with red-shifted opsins (such as Chrmine at 635 nm [Chen2021]) to reduce beam broadening in tissue. Switching the operating wavelength from 470 nm to

635 nm can extend the scattering length by 30-40% \cite{Moy2015,Mesradi2013}. Second, the steering range of the OPAs can be further increased while maintaining beam divergence by (1) reducing the phased array pitch and (2) increasing the number of emitters in the phased array to preserve the aperture size. For instance, OPA Type IV is expected to gain a 1.8-fold increase in steering range by reducing the phased array pitch from 500 nm to 300 nm based on the OPA steering range equation in \cite{Liu2022}. As the number of emitters increases, thicker waveguides or higher index waveguide materials can be used to minimize the increase in OPA footprint, as discussed in Supplementary Section \ref{sup:scalingdesign}. Finally, future OPA designs could adopt out-of-plane focusing grating strategies demonstrated in \cite{xue2024implantable, Shirao2022} on the slab grating or introduce a parabolic phase shift at the delay lines \cite{Hosseini2024} to generate steerable focused beams, thereby reducing beam broadening in tissue and localizing the stimulation near the focal point where the intensity is the highest. \cite{xue2024implantable} has demonstrated that it is possible to achieve focused beams with widths $< 10 \text{ \textmu m}$ located 50 \textmu m from the probe in tissue with grating-based emitters

Adding phase modulation to the OPA device brings two benefits. First, it allows more degrees of control on beam shaping as proposed in the Supplementary Figure 12 of Ref.22, generating different beam patterns in tissue beyond steering (i.e., focused beam at different depths). Second, as mentioned, it opens up possible paths to compensate for scattering-induced phase distortions. At the same time, we agree with the reviewer that implementing guidestar-free feedback *in vivo* to optimize beam width can be challenging especially since our probe is not equipped with an image sensor for detecting reflected fluorescence signals like other microscope setup [R1, R2]. Nevertheless, as the ultimate goal for our device is to stimulate selective neurons while minimizing stimulation effects on surrounding neurons, it would be interesting to explore the use of neuronal firing rate as a feedback signal for optimizing the incident power on selected neurons through wavefront shaping. Our *in vivo* results in Fig 5d also show that the firing rate of the stimulated sorted spikes depends on the optical power of the beam. One downside of this approach is that spiking activity has a slow temporal resolution of $>1 \text{ ms}$ and typically it would require multiple stimulation pulses to obtain stable average firing rate responses from the neurons. Concurrently, the dynamic scattering in live tissue can lead to unstable phase compensation over time [R3]. Future research is needed to investigate these two factors within our targeted stimulation depth ($<140 \text{ \textmu m}$).

We have added the following details in the Discussion to clarify the possible benefits of having active thermal phase shifters beyond implementing adaptive optics and using firing rate of the recorded neurons as the feedback signal to optimize the beam selectivity.

A long-term solution could involve implementing active thermal phase shifters for each waveguide emitter in the end-fire phased array. This approach would (1) provide greater control on beam shaping beyond beam steering along an arc (i.e., adjusting beam focusing depth \cite{Mohanty2020}) and (2) enable phase optimization to compensate for scattering-induced phase distortions \cite{Filip2024}. Firing rate of the targeted/untargeted neurons could be explored as the feedback signal to optimize the beam selectivity on stimulating neurons, as the firing rate is dependent on the incident beam intensity as seen in Fig. \ref{fig:fig5_invivo}d).

Reviewer #2 (Remarks to the Author):

Referee Report on the Revised Manuscript:

This paper introduces a neural probe design with a grating-based light emitter and an integrated optical phased array for steerable optogenetic stimulation. It demonstrates beam steering in transparent media but falls short in scattering media. I have reviewed the revised version of the manuscript, and I am pleased to note that the authors have satisfactorily addressed the comments raised in the initial review. The revisions have significantly improved the clarity and overall quality of the manuscript, and I believe that the work can be recommended for publication in Nature Comm. Engg. I appreciate the additional discussion provided in the supplementary section, particularly the section regarding scaling constraints and design trade-offs. This discussion adds significant value to the work, making it much more comprehensive and thorough. The insights on how these constraints affect design decisions are particularly useful for the audience seeking design guidelines for photonic devices for biomedical applications.

We thank the reviewer for recognizing the improvements we made to the work.

I have a few minor but important comments that should be addressed to further strengthen the manuscript.

1. It would be useful to provide citations/sources for the range of refractive index considered for PECVD and LPCVD SiN (Lines #106 - 109 in the manuscript)

We thank the reviewer for the suggestion. We have now added citations (Ref.24 , Ref.36) to provide references for the typical refractive index values we considered for PECVD and LPCVD SiN. The refractive index of these films is process-dependent and can vary based on the fabrication methods used by the foundry. For completeness, we included a range of refractive indices for PECVD SiN in our manuscript, because measurements from our recently fabricated wafers at Advanced Micro Foundry showed higher values (up to $n = 1.9$) than those previously reported in Ref.24.

2. In the in vivo experiment section, it is stated: "The laser scanning system used to address the OPA emitters, could turn on one OPA at a time, with a switching speed of about 5-10 ms."

I would appreciate it if the authors could provide more detail on whether this switching speed is sufficient for typical optogenetic experiments. If not, it would be helpful to suggest hardware or system-level designs that could enable faster switching or even concurrent use of multiple OPAs for stimulation

Thank you for your question. Our current switching speed is sufficient for neural mapping experiments that do not require sub-millisecond level precision, such as the experiments described in [Ref.1,Ref.10, Ref.41]. However, ideally, to permit photostimulation at the temporal resolution of a single action potential, the switching speed between OPA emitters should be less than 1 ms (> 1 KHz) [Ref. 40]. To achieve faster switching, we could design an integrated thermo-optic switch network directly on the neural probe. This would allow rapid switching between OPAs at a frequency of ~ 50 KHz, replacing the free-space optics in our current system, as described in [Ref.22]. Additionally, this integrated network would enable simultaneous use of multiple OPAs for stimulation.

We have included this improvement in the manuscript,

“The laser scanning system used to address the OPA emitters could turn on one OPA at a time, with a switching speed of 5 -- 10 ms. This speed is adequate for some applications of circuit mapping \cite{Pisanello2017,Gill2020,Wang2007}. However, to permit neural stimulation at the temporal resolution of a single action potential, switching speeds should be below ≈ 1 ms \cite{Grossman2010}. Future optimizations could include the design of an integrated thermo-optic switch network on the neural probe, allowing OPA switching at ≈ 50 KHz and enabling concurrent use of multiple OPA emitters \cite{Mohanty2020}.”

3. In the "Beam Profile Characterization" section, the authors mention: "The tunable laser source has a response time of 55 ms with a 1 nm step size, resulting in a tuning speed of approximately 1.1 seconds across the full FSR."

Could the authors comment on whether this tuning speed is sufficient for the intended/envisioned experiments using the neural probe? If not, what optimizations or modifications might be considered to improve the speed to meet the experimental demands?

Similar to question 2, to achieve an optimal temporal resolution in optogenetic experiments the wavelength tuning speed should be less than or equal to the time needed to trigger a neural response (~ 1 ms) [Ref. 40]. Nevertheless, experiments employing scan-based schemes in free space for one-photon excitation of ChR2, have shown that a 100 ms interval between consecutive photostimulation events can be sufficient for mapping neural connectivity in the cortex [Ref.1].

Since the current system's response time (55 ms) is faster than the 100 ms inter-location interval used in some previous studies, it may be sufficient for neural mapping experiments where ultra-fast timing is not essential. However, for experiments requiring higher temporal precision [Ref.22, Ref. 40, Ref.41], we could consider enhancing the wavelength tuning speed of the laser source—comprised of a supercontinuum white laser and a diffraction-grating-based tunable bandpass filter—by replacing the mechanically moving diffraction grating used for wavelength selection with an RF signal-controlled acousto-optic tunable filter that enables sub-millisecond wavelength switching [Ref. 20, Ref 42].

We have included these details in the text,

“The tunable laser source has a response time of 55 ms with a 1 nm step size, resulting in a tuning speed of approximately 1.1 seconds across the full FSR. Although this scanning speed may be sufficient for some neural mapping experiments \cite{Wang 2007}, experiments requiring higher temporal precision \cite{Grossman2010, Mohanty2020, Gill2020}, could benefit from replacing the mechanically moving diffraction grating in the tunable bandpass filter with an acousto-optic filter, to enable sub-millisecond wavelength switching \cite{Misono1996, Segev2016}.”

Overall, the manuscript presents notable advancements in the field, and with these minor revisions, I believe it will be even more impactful.

We thank the reviewers again for the support on the paper and the valuable feedback that helps to improve the quality of the work.

Additional Edits

- Light edits the abstract to meet the 150 word count limit
- Added a title sentence to each Figure

References

[R1] Antoine Boniface, Baptiste Blochet, Jonathan Dong, and Sylvain Gigan, "Noninvasive light focusing in scattering media using speckle variance optimization," *Optica* 6, 1381-1385 (2019)

[R2] Aizik, D., Levin, A. Non-invasive and noise-robust light focusing using confocal wavefront shaping. *Nat Commun* 15, 5575 (2024).

[R3]B. Blochet, L. Bourdieu, and S. Gigan, "Focusing light through dynamical samples using fast continuous wavefront optimization," *Opt. Lett.* 42, 4994-4997 (2017)

Reviewer #1 (Remarks to the Author):

Thanks for addressing the comments and clarifying the points in the manuscript.

The authors have addressed my comments in a satisfactory way.

Reviewer #2 (Remarks to the Author):

The manuscript presents an integrated photonic phased array for in vivo beam steering, enabling spatially controllable optogenetic stimulation. The authors have satisfactorily addressed all the minor comments raised in my previous review. The added clarifications, along with the inclusion of relevant references, have improved the manuscript's clarity and contextual understanding of the work.

At this point, I do not have any further comments or questions for the authors, and in my opinion, the manuscript can be considered for publication.

We thank both reviewers' feedback and recommendations for publication in Communications Engineering.